# Evolution and Management of Illegal Settlements in Mid-Sized Towns. The Case of Sierra de Santa Bárbara (Plasencia, Spain)

Víctor Jiménez Barrado

Instituto de Geografía, Pontificia Universidad Católica de Chile, Santiago de Chile 7810000, Chile; victor.jimenez@uc.cl; Tel.: +56-2-2354-4755

**Abstract:** The illegal urbanization of rural areas near cities has unveiled failures in urban management. In many cases, urban policies have ignored this fact until the spaces have consolidated. This is the example of the Sierra de Santa Bárbara (Plasencia, Spain), where legalization becomes one of the most feasible solutions. The present work analyses its residential evolution during the last four decades through historical orthophotos review. Along with this, it evaluates public–private conflicts (homeowners vs municipal government) using regional newspaper archives. The results indicate that the strategy of ignoring illegal development increases these problems, leading to legalization as the only possible urban policy. In conclusion, the administration's response is delayed and forced by critical consequences, which prevents learning in urban policies and new solutions that join legality and sustainability.

**Keywords:** illegal urbanization; informality; informal settlements; legalization policies; rurbanization; urban policies

## 1. Introduction

The literature about informal settlements focuses primarily on land grabs and on the creation of slums, which are a common phenomenon in developing countries and even in developed ones, some of them included in the Global North [1]. Then, this type of informality is associated with marginality [2–4]. Even the UN [5] agrees with this idea in its New Urban Agenda.

In underdeveloped spaces, urban policy debate centers on its physical and social precariousness [6] and on disputes over land tenure [7]. There, irregularity starts with the illegal occupation of land and continues through construction. Informality is at the root of the right to housing [8,9]. In these spaces, the lack of social justice contributes to prioritizing legitimacy over legality [10] and to causing governments to adopt a mediating role to solve social struggles [11]. As a result, various mechanisms are used to make both concepts comparable to ensure the abovementioned right to housing [12,13].

In more developed contexts, where the right to housing is ensured, informal settlements do not relate to housing needs but rather to a change in lifestyle and the acquisition of second homes by the middle class, in which leisure time plays an important role [14,15]. In these cases, the urban policies debate about informality focuses on other issues such as environmental impacts and legal consequences.

This is the main reason for defining informal settlements as illegal settlements when they occur in developed contexts. While the term "informal" involves satisfying existential human needs [16], the term "illegal" focuses on this way as one more option and not as an irremediable destination. In other words, informality can be treated as a particular aspect of noncompliance with the law [17]. By means of this denomination, it is intended to clearly define the difference between what some authors [18] name as necessary illegal building and speculative illegal building.

Against the logic of solidarity, this type of urbanization has not been closely studied and has become more accepted by governments [19,20]. If this behavior continues, there is a risk of considering illegal urbanization as the model rather than a negative symptom or a dangerous consequence of it [21]. This kind of development is based on two trends. The first one is a centrifugal one, triggered by the idea of rejection of the city [22], and the second one is a centripetal one, based on the chlorophyll (go green) idea or culture [23,24]. In this way, the urban lifestyle is replacing the rural one [25] and altering its landscape. While one movement expels population, the other directs it to a destination, normally creating a new low-density residential area. It is a common trend taking place during recent decades in the Mediterranean Europe [26], including Spain.

In addition, the Mediterranean basin presents particularities such as the role of the massive middle-class that illegally builds on its own plots (mostly secondary homes), a phenomenon shared in countries like Italy and Greece, among others [27,28]. For this reason, more attention needs to be paid to illegal urbanization, as its magnitude can be significantly increased under this context. As a matter of fact, in Spain, this has also been a common problem, which has initiated urban policies in a lot of its autonomous communities [29].

In that region, the difference between legal and illegal is clearly justified by an environmental issue: the relevance of that process is marked by the high vulnerability of the Mediterranean context related to climate change [30–32]. This global meteorological phenomenon creates new deficits in the region [33], and these are aggravated by a change in the urban planning model. The traditional Mediterranean model is considered one of the most sustainable ones because of its compactness [34,35]. Therefore, implementing a less efficient model (such as one with scattered settlements) in this vulnerable geographical context increases the environmental footprint of urbanization and the pressure on the physical environment [36].

The impacts are soil erosion and degradation [37], massive catchment of water and pollution of scarce water resources [38], loss of native vegetation due to changes in land use [39], increased air emissions due to raising mobility [40], and risk management [41–44]. Therefore, dispersed illegal settlements are deeply modifying the physical elements of the landscape.

Then, urban policies can follow three ways: ignore illegal settlements, integrate them (or not), or use them to improve planning guidelines [45]. In Spain, the first option has been the favorite over decades. As in other contexts [46], the political influence of homeowners (by socioeconomic status or number of voters) has been crucial to tolerate illegal developments. This has led to large illegal urbanizations and highly complex land planning problems. As a result, several regional governments such as Extremadura's government (urban planning is dealt by autonomous communities in Spain), where Plasencia is located, have been pursuing the second way [47–49].

This problem is dealt with by a flexibilization of urban laws and legal instruments, even though these kinds of settlement are not automatically legalized [50]. These processes should be promoted on a smaller scale, both by local governments and by homeowners, and supervised by regional governments. Modifications to the Master Plans (MP) are unavoidable.

The legalization of housing units opens a legal, political, and social debate. The results of this process can be ambiguous, bringing positive and negative consequences [51]. Dealing with formal urbanizations in legal terms and dealing (or not dealing) with informal and illegal urbanizations evince parallel management (that is, two ways of doing that never meet and cross) [52], but the legalization processes make them clash. This confrontation creates winners and losers that vary from case to case. Thus, it can provide a boost to the formal economy, raising fiscal revenues and contributing to job creation without forgetting the assurance of sustainable development [53]. By contrast, outside of contexts marked by social segregation, the result of these policies can create feelings of resentment among the middle class as there may be a benefit for a few at the expense of law-abiding citizens [54].

Therefore, the dilemma is not just about the convenience of legalization but how to carry it out. These processes involve adding economic worth to the value of use, since only after acquiring legality the market value of buildings is fully in use. Consequently, this facilitates the production

and reproduction of the model. For example, in neoliberal countries, this process is quite easy as public lands are regularized more quickly [55]. The commercialization of these houses distorts the market [56] and, along with it, the objectives of the planning. Furthermore, when legalization processes do not require enough urbanization and environmental standards, the resulting urban centers and their impact on the environment do not disappear even though they reach legality. In fact, regularization can mask an ambiguous political action under a technical exercise [57]. A serious consequence could arise, in which the opposite goal is achieved: reaching legitimacy through legality.

Currently, public administrations are required by law (for urban and environmental purposes) to choose the second way, with two clear alternatives: legalization or demolition. As a result, governments and private entities (homeowners and owners 'associations) have become enemies. Then, dialogue is difficult and only takes place in extreme situations required by judicial sentences. Until these come out, the two options are so traumatic that they generate paralysis and the return to the first way.

A case study about a consolidated illegal settlement in the Sierra de Santa Bárbara—SSB—(Plasencia, Extremadura) is presented in this paper. In this space, the first way of urban policies [45] had been operating for four decades; however, it is not an option any longer.

In the study area, demolition orders have triggered the second way in terms of legalization (promoted by the owners), which requires the local government to respond in an affirmative or negative way. The aim of this work is to demonstrate that the strategy of ignoring the problem (first way in urban policies) causes its increase, just as the second way is a specific solution in space and time, activated only in extreme cases for the competent government (conflict social, legal problems, or media attention). Therefore, this work's evidence of the first way is an erroneous urban policy and the second way should only be a good option if it represents a path to the third way in the future (learning and planning improvement).

For this reason, data about urban development and evaluation of the administration interaction with illegal settlements are provided. Also, there is an examination of the regularization proposal viability.

Finally, based on the main results (existence of a high level of landscape transformation, indolence of the municipal government, and proposal insufficiencies presented by the owners) and on the existing legal conditions, it can be deduced that legalization is the most feasible urban policy but not the only possible policy in future for the study area. Hence, this work presents some proposals to overcome the current situation standstill and to start exploring the third way, which could lead to a better management of illegal settlements and a new role for governments.

## 2. Materials and Methods

### 2.1. Evolution and Characterization of the Study Area

The first step within the first block is to describe the study area in urban terms. To do this, I have worked with the MP land classification plans of Plasencia. These were found in the Land Information System of the Government of Extremadura (SITEX) [58] in JPEG format without georeferencing. Those with a larger scale of 1:10,000 have been chosen to obtain a more detail picture. Locations have been determined using ArcMap 10.5 (ESRI, Redlands, CA, USA) based on the most recent orthophotos (year 2016). This free cartography can be found in the National Center of Geographic Information. In this process, the control points are precise enclaves (corners of built elements) located at different points distributed in various areas in the map. The overlapping between urban plans and orthophotos, both georeferenced, has allowed to vectorize the land classes (with polygon topology). Each of these classes has different building conditions.

The second step is detecting the existing buildings within the study area. This is a crucial step as most buildings found were illegally constructed and are not officially registered. Visual scanning was then carried out (east–west direction, from north to south) over orthophotos at a maximum

possible scale of 1:600. Vector layers of points and polygons are developed for precising number, area, and location of houses.

The third step involves the classification of buildings. Two types are created: housing units and other building structures. To determine each use, field work was carried out with aerial photography support through drones. This work can be easily done because the dimension of the surrounded field is quite small (25 km$^2$). Because the field size is small, residential land uses are identified through the high precision method and no other and more complex methodologies are needed to detect urban structures [59,60]. In previous occasions [61], for larger field studies, detections are carried out over the specific orthophotography (using urban-related structures consistent with residential uses, such as swimming pools or gardens) or Google Street View images.

The fourth step is to determine the date of housing units. This is possible using various older aerial photos and orthophotos. To facilitate the process, I have worked with a Web Map Server version (Ministry of Development, Madrid, Spain). Six dates are available: 1984, 1998, 2002, 2005, 2012, and 2016 (in which digitalization is carried out). The vector layer is contrasted with the available images and crosschecked, from recent to older series, and thus, the existence of urban structures is analyzed.

The result shows a layer containing the urban land classification and another layer showing the dates of housing units. This allows for the overlapping of another layer (in a shapefile georeferenced format) containing "urban authorizations" given by the autonomous government. From examination, it was then possible to find out whether the housing units conform to the required regulation. If they do not, they are considered illegal. Briefly anticipating the results, there is no legal document (urban authorization) approving the construction of any buildings in the study area.

## 2.2. Analysis of Local Government Performance and Urban Policies Used for Illegality

The second part is used to analyze public policies implemented in the study area. The documents examined were divided in two parts. The first contains the private proposal for modification of the MP carried out by the owners and the environmental impact assessment study. Also, the urban regulation sections of the MP, its Environmental Sustainability Report, and the MP environmental report carried out by the regional government were examined. All these documents are available accessing the regional government website "Extremambiente" [62] and the abovementioned sources.

Firstly, the layer containing illegal settlements areas (extracted from the MP environmental report) was overlapped (the proposal to legalize them is based on it). These areas are officially named "Illegal Urban Settlements under Regularization" (RUCI). The latter layer is obtained following the same procedure described in the first block, and it adds to the former layer of urban land classification. Through this, the perimeter and the number of dwellings that benefit from the proposal have been determined as well as its relationship with the different land classes, with special attention to those included in the landscape protection of the SSB. Results and methods can be compared to other ones such as the one used to reclassify land to Legalizator Developable Land (LDL).

All documents are studied separately but with cartographic support. In addition, an analysis to detect conformity to urban and environmental regulations was carried out. A key point to verify document validity was checking their conformity to regional legislation.

A review of the regional press was carried out to examine the relationship between the local government and the illegal settlement. This source is one of the most reliable and trustworthy that could be found to check its evolution through time [63]. The SSB illegal settlement is a high-profile case in the region; therefore, any new development is registered in the local press. Also, a search on the digital newspaper archive of the two most widely read newspapers was carried out to examine relevant news about the study area ("Diario Hoy" and "El Periódico Extremadura"). Of particular interest were the sections on political statements, legal news, and owners' associations.

## 3. Results

*3.1. Study Area Characterization and Residential Evolution over Non-Developable Land (NDL)*

The SSB corresponds to a minor mountainous area (669 m high) from NE to SW located between the city councils of Malpartida de Plasencia and Plasencia (Figure 1). The principal urban center of the latter (360–430 m) is a mid-sized town (30,913 inhabitants) located in the central-west side of the Iberian Peninsula. Within its closest geographical area, Plasencia has historically occupied an important role. Before the provincial division of Spain in 1833, its demographic differences with Cáceres (current capital city) were minor. In 1797 [64], Plasencia had 4852 inhabitants and Cáceres had 6860 inhabitants. Since then, the concentration of administrative services in the capital city Plasencia has prejudice, although it has maintained its influence throughout the north of the province. Furthermore, the growth of its territorial relevance has been limited by the influence of other provincial capitals (such as Salamanca) or large cities in other provinces (Talavera de la Reina, province of Toledo) as well as by the proximity of the national border between Spain and Portugal.

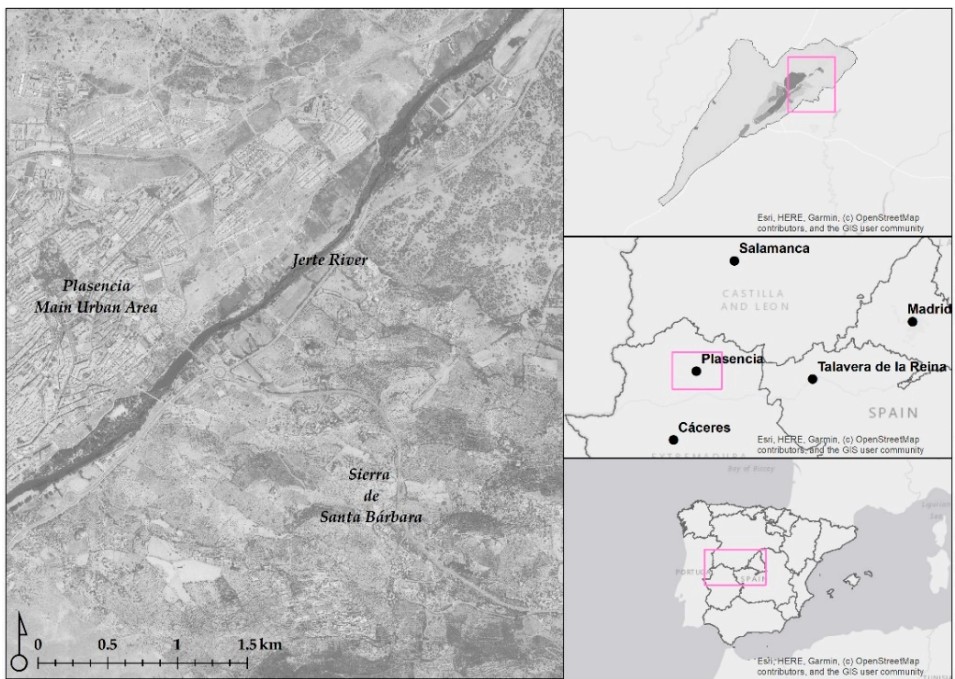

**Figure 1.** Localization of the study area. Source: Own elaboration.

Despite its current size, Plasencia is the fourth most populous city at the regional level and the second one at the provincial level. In fact, it is regarded as a main commercial and service-related center within the northern sector of the autonomous community of Extremadura, the region in which small towns predominate.

The study area has been urbanized under these influences. In this case, the number of housing units within the NDL reaches 464, including the lowest areas and closer zones to the Jerte River (this watercourse constitutes a natural division between the city and the study area). The average floor plant size of dwellings is 149.45 m$^2$ (69.68% exceeds 90 m$^2$ and 24.86% is above 180 m$^2$), with a very high prevalence, but not specified in this work, of single-storey houses (upper two-storey houses do not exist). Occupation of this territory has taken place traditionally over the years due to its highest elevation and its role as a viewpoint over the city amongst a landscape of outstanding beauty. The attractiveness of the study area lies mainly in the possibility of acquiring or building a dwelling within a natural environment but close to an urban center. In addition, they are more spacious and comfortable housing typologies at a lower cost because the homeowners save the costs associated with

legal housing (taxes and legal payments to construction professionals). The land ownership structure is highly fragmented, so the average area of land properties is 1.33 ha (it does not reach the 1.5 ha required by law to build a home in NDL). In fact, under these conditions are 85.56% of the plots built in the study area. Along with this, judicial boundaries of the plots and those observed in the orthophoto do not coincide, which means that the largest plots have been illegally divided and built.

Within an analysis of its building evolution, 25% of the total housing units (total of 118 units) had already been built in 1984. This number has been steadily rising (Figure 2). In fact, the record for the period 1984–1998 (173 housing units built) shows a figure very similar to that of the following 14-year period (167) despite the years between 1998 and 2012 establishing one of the most dynamic economic and urban development periods in the country's history [65].

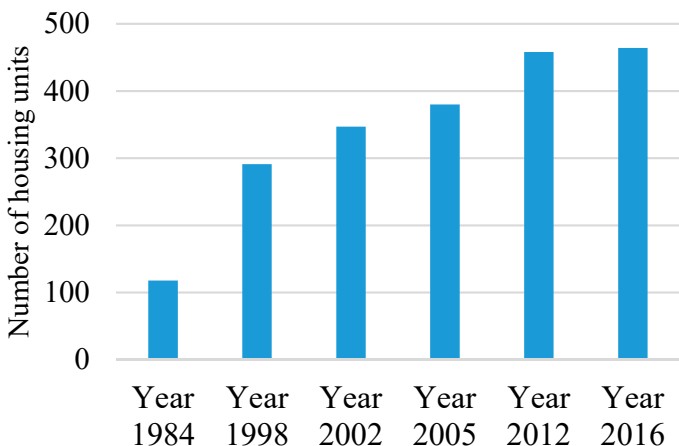

**Figure 2.** Evolution of the total number of housing units in non-developable land of Sierra de Santa Bárbara. Source: Own elaboration.

From the beginning, housing units occupied the entire mountains' slopes, from its lowest point next to the Jerte River (340 m high) to its highest areas (650 m high). Over time, certain areas have become denser. This is a zone of high slopes, previously occupied by xerophytic vegetation; part of it has remained. In addition to the scattered dwellings, the landscape is dominated by olive trees and other dry crops arranged on terraces as well as artificially irrigated spaces such as gardens and orchards. Here, the number of swimming pools (300) is abundant in a context of summer water shortages. Land plots are irregular and atomized, delimited by metal fences and roads (the main ones are paved, but secondary ones are not).

Currently, according to Plasencia's MP (approved in 2015), the study area is split into 3 zones, divided into two land classes. In the lower part (the one closest to the river and the city, with lower slopes), there is land for development (for a legalization purpose)—LDL—while the higher areas are classified as NDL. The latter is divided into categories, so that the lower half of the slope (up to 360–420 m) corresponds to the "common NDL", and the upper part corresponds to the "protected NDL" (>420 m), in this case, a landscape protection type (Figure 3). It is in these two categories that the current proposals for housing legalization in Plasencia applies.

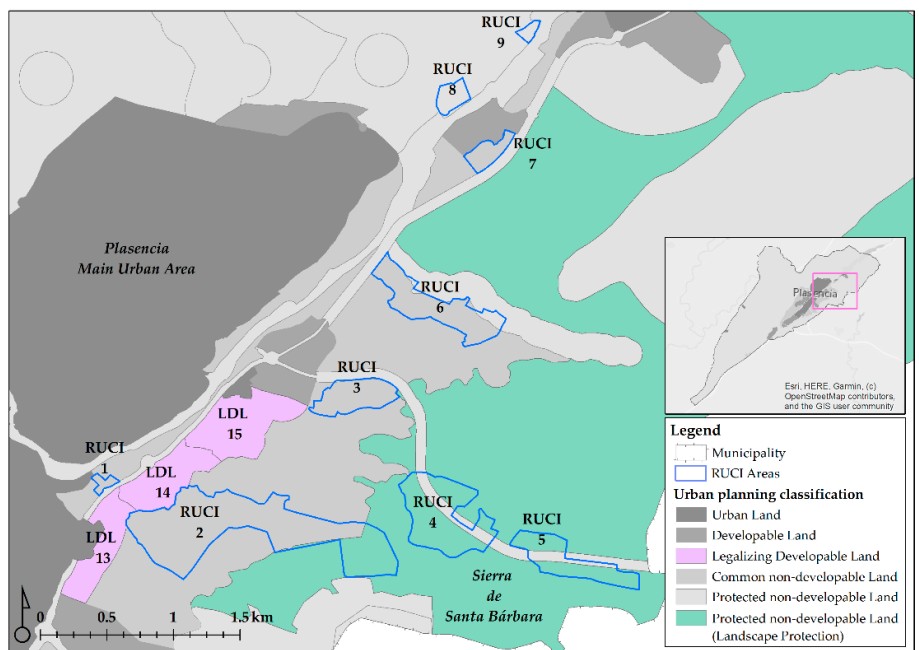

**Figure 3.** Location of RUCI (Illegal Urban Settlements under Regularization) over the urban land use classification plan. Source: Plasencia Master Plan (MP), own elaboration.

### 3.2. Measures and Attempts of Urban Legalization

Uneven classification is an initial way of carrying out urban policies. There are two options. The first one is represented by the LDL classification, a procedure used only in 17 municipalities in the region [66]. This one anticipates a result: housing legalization at the planning stage and a higher density urban development. The second one, corresponding to NDL, is more imprecise: all possibilities are open for illegal housing, from legalization to demolition.

#### 3.2.1. LDL, an Irreversible Urban Integration

In Plasencia, the first option originated from the previous MP (approved in 1997). The current MP keeps this structure and divides the land into three areas (LDL 1, 2, and 3), which have a total dimension of 643,649 m$^2$. These areas are located on the eastern bank of the Jerte River, at the bottom of the SSB slope. Each area will be developed individually according to the use and building conditions (Figure 4). This requires the approval of a partial management plan that would define in detail land uses (previously and generally described in the MP).

This space is currently occupied by 45 dwellings, mostly built when the space was still considered NDL. The construction processes began more than four decades ago, giving rise to the urbanization of the entire study area. The highest building rate was achieved between the 1980s and the mid-1990s. Since its incorporation to the LDL, the construction rate has slowed down (only 4 dwellings). This has occurred simultaneously with the abovementioned period of great dynamism.

Despite being a straightforward procedure, deadlines have not been met (they were due in 2019), so homeowners are once again in a legal limbo. Preexisting buildings affect and hinder new developments. This is because legal inability to demolish housing prevents projected development of the MP.

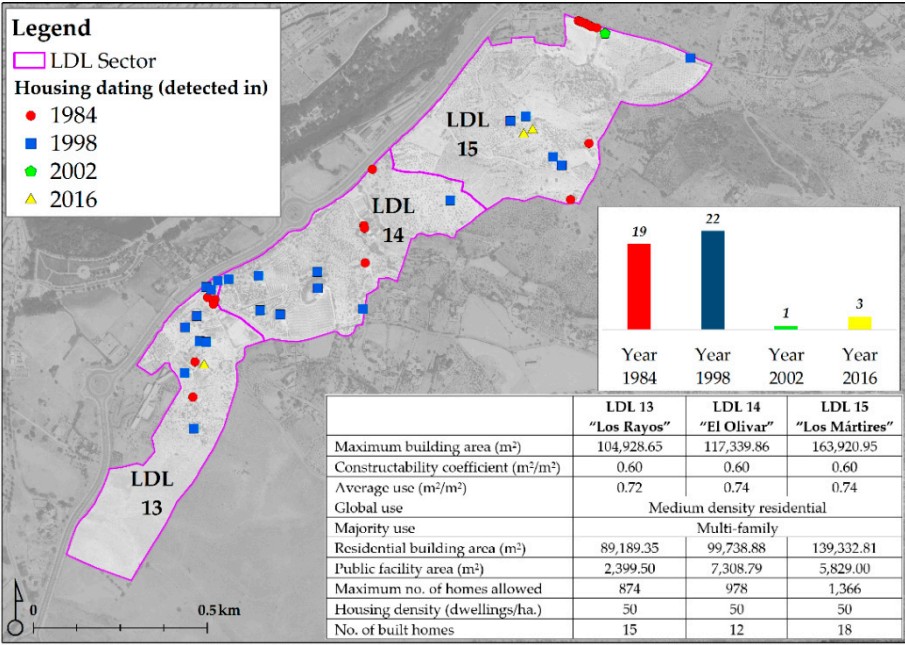

**Figure 4.** Dates of housing units in Legalizator Developable Land (LDL) sectors. Source: own elaboration.

### 3.2.2. Regulatory Proposal in the NDL and Its Feasibility

The potential second way, in NDL, will require a MP´s modification to be successful. The current Environmental Sustainability Report states that good location, low cost, and speculative behavior has caused illegal development in the study area. This document delimits 9 illegal residential zones but limits the viability of regularization to those located in "common NDL". However, it states that legalization of RUCI 1 will be difficult as this area is located in a flood risk zone. Additionally, it explains the difficulty of legalizing housing in the "protected NDL" since they generate landscape impact (particularly in RUCI 2). Therefore, the Environmental Sustainability Report does not include a concrete solution.

This lack of concretion is criticized by the MP environmental report (of regional competence). In another line of criticism are homeowners in the "landscape protected UDL" (RUCI 2, 4, and 5). From their point of view, the impact on the landscape caused by their homes is less than officially calculated. Based on these criteria, the 2017 proposal goal is to initiate a legalization process (allowed by the MP's environmental report) including their homes as well. Demolition orders on some homes and the fear of new orders have advanced this process.

Homeowners rely on the fact that Environmental Sustainability Report recognizes "the necessary compatibility between urban uses and the conservation of natural, landscape, and cultural values" of the SSB and that this will require "regulating certain consolidated urban areas or even tolerating some type of new developments". Therefore, they propose 3 choices, although only one of them is the actual proposal:

(a)　Allow legalization through RUCI areas in the "landscape protection NDL".
(b)　Allow legalization through RUCI areas over the entire NDL.
(c)　Change the "landscape protection NDL" classification to "common NDL".

The document itself criticizes the last two choices, considering that proposal (b) would further worsen the problem by legalizing housing buildings throughout the municipality while proposal (c) would contribute to a greater transformation in the area. While proposal (b) may have a legal basis by relying on legal technicalities, proposal (c) has no chance because environmental values exist in this space.

Neighbors therefore present proposal (a) as the best possible option. They argue that the MP is more restrictive than the law, and furthermore, it does not propose solutions. According to their point of view, housing legalization in this place will not have a greater impact (neither by land occupation, by water spills, nor on vegetation or landscape), since it is a spatially confined area and regulates existing urban development while restricting new developments. Following this premise, landscape impact is presumed as low or nil.

However, this argument does not consider that urban development of the SSB (Figure 5) and its current state already have a real impact on landscape. Previously, the MP environmental report criticized it for not proposing impact mitigation or correction measures. Corresponding to its own criteria, the regional government will also criticize this owners' proposal and their denial of landscape impact.

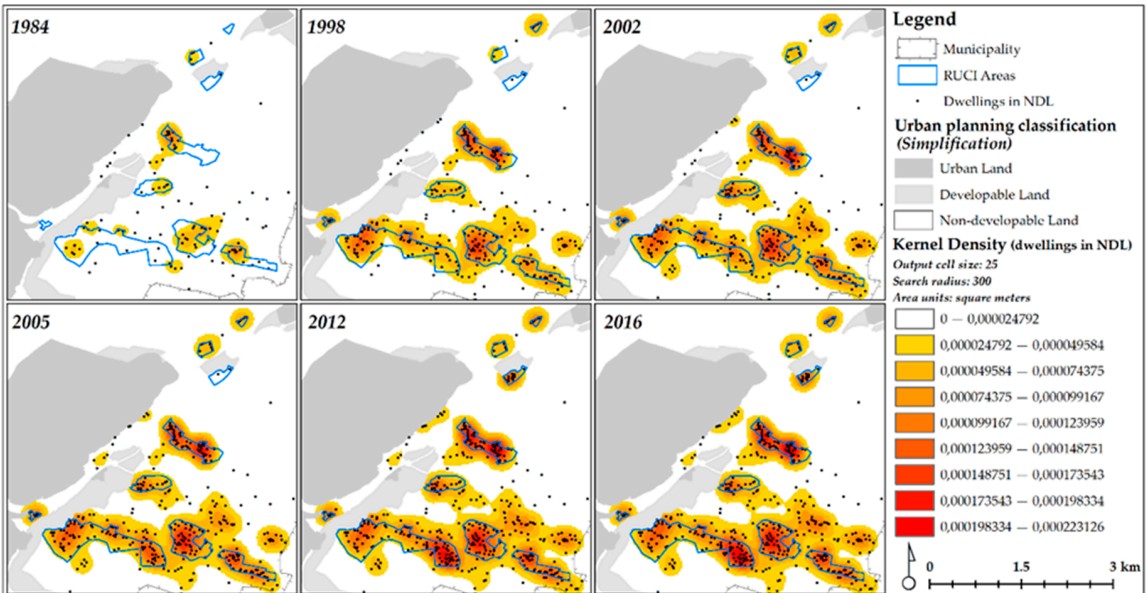

**Figure 5.** Evolution of homes built in the non-developable land of the Sierra de Santa Bárbara (SSB). Source: Own elaboration.

Without regional approval, action is not viable. According to law, an environmental report must be approved by regional government in order to modify the existing MP. However, this authorization is difficult because building dwellings within natural areas involves threats. The houses do not have all the services and infrastructures of urban land (garbage collection service, web connection, urban equipment, etc.), and therefore, they are potential sources and victims of socio-natural risks and various types of pollution.

Despite this, its legalization must occur in one form or another. Most homes will not disappear because the criminal code prevents their demolition when the statute of limitations elapsed. If there is no solution, only a few houses will disappear and the rest will remain under the current status quo and thus affected by corresponding consequences.

### 3.3. The Municipal (City Council) Point of View Regarding Illegality

A review of the regional press shows that SSB urbanization has long been known to local government and society [67,68]. Its media coverage is due to legal problems of local politicians (complaints and convictions for malfeasance) and homeowners (demolition orders, carried out or not). The media interest is reactivated in specific periods when planning initiatives, such as the creation of a new MP or the legalization proposal in 2017.

The local governments' position has been changing with time. First, there was total ignorance of the fact, then underestimation of the problem, and finally attempts to manage it. Thus, until the

end of the 20th century, there were no press reports on the urbanization of the SSB and no reactions of local government officials. In the first statement made by the mayor in relation to the problem (1997) [67], it was evident that municipal inaction was a local policy strategy for at least the last 17 years. His political actions can be summarized in his own statement: "I don't think this can be an issue for the city; what worries me, is that construction, which is the spark of local economic activity, continues to be carried out".

In the 21st century, the press shows a greater awareness of local government over SSB issues and local politicians reveal their excuses for not acting. In 2005 [69], the Social Welfare councilor acknowledged that there is no action on the area because "it has a huge political cost", while in 2009, the city planning councilor blamed it on the "lack of personnel" [70]. A year earlier, that same councilor was calling for community responsibility and asking for "citizen discipline" [71].

The analysis of the press confirms that the SSB is a space of conflicting interests. On one hand, there is the homeowners' interest, for which attitude has also evolved. Their first complaint was related to improvement in urban conditions, mainly about their connection with the main urban center. This request was denied by regional and local governments, recalling the previous period when the problem was ignored [72]: "There are no housing units in the SSB, because—the regional government—consider they are illegal". After this, the homeowners' claims have focused on housing legalization.

On the other hand, environmental groups are against homeowners' interest [73], while the professional association of architects has been advocating since 1995 for better regulation in this place that would avoid further "comparative issues" [74]. Nevertheless, there are projects such as the implementation of a wind farm that produces a common negative answer by all parts for its visual, noise, and environmental impact in the area [75,76].

Since the beginning of the 21st century, the local police have reported illegal construction of houses and publicized it. Annual press reports, detailed for the SSB, have come out in the newspapers since 2003 [71,77,78]. This work has contributed to changing the municipal attitude. Local government's claims have increased as shown by the urban planning department chief's declaration: "Before, we used to turn a blind eye, but now, as soon as these infringements are detected, files are released and sent to the Public Prosecutor's Office" [79]. According to him, this change is taking place to "ensure the principle of equal rights amongst citizens", taking this case as "a matter of justice" [80].

Since 2009, when the MP began its discussion process, the press started writing about the possibility of legalizing homes in this geographic location. This action has always been considered by both the local and regional governments as a solution for dwellings groups [81] but never as a solution for single housing units. From that moment, the administration began to send double messages to the public: the costs of legalization will be borne by the owners [82] and the battle against illegal housing will continue. However, this message has turned out to be contradictory, as the numerous demolition orders are not being carried out either before [83] or after this statement [84].

What has really triggered the concrete possibility of legalization has been the legal proceedings from 2017 onwards against political leaders of all parties, both conservative and progressive [85–87]. This situation, together with the political battles and political leaders fears of possible convictions, have generated serious actions never seen before in the administration, such as an increase of complaints to homeowners [88] and the first demolition of an illegal dwelling in the area of study on September 17, 2019 [89]. Also, this meant the creation of the homeowners' association in the area and the first legalization proposal. All this has increased even more the media coverage. Up to that year, annual press reports on this place were below ten, while in 2017 and 2018, they have widely exceeded a hundred.

As a consequence of these critical events, the local government is acting today to ensure legal security. This causes legalization processes to slow down while the homeowners demand a quick response [90]. The last mayor's statement about illegal housing on the study area in January 2020 was as follows: dwellings on SSB "have been going on for 40 years and we can't solve this issue in

a few months, not even in a few years" [91]. The city council has already started the administrative procedure (February 2020) without knowing faithfully when and how it will end [92].

## 4. Discussion

Contrary to the fact that legalization policies should focus on urban regulations and standards [93], the results of this research show that the main issue to be addressed is to keep an invariant political discourse and the predominant role in the public administrations during the process. The latter has not occurred in the study area, and the consequence has been a growing illegal urbanization and a process still unsolved despite clear rules in urban legislation.

The success of the legalization processes could be considered relatively easy in physical terms (provide infrastructures and public urban services). There are examples [94–96] that show the combination of top-down and bottom-up urban policies as the best way to solve material deficits. While the public part contributes the financing, the private part proposes collective ideas. There is a great consensus about the idea of participatory urban planning that considers "the others" as the best strategy [97,98].

However, these examples work in a vulnerability scenario with strong social and political pressure (justified by humanitarian issues) [99] but not in one in which the middle class participates and the basic needs are covered. In the case studied, there is no claim based on social need but on private benefit. The legalization proposal studied seems to indicate that losing interest and initiative in terms of urban planning discipline by public administrations may cause the union of the homeowners and hence trigger a social pressure increase.

As has been stated for other contexts [100,101], the adoption of a position that is not clearly dominant by public administrations can lead to an escalation in illegal urbanization. For this reason, it is important to point out that the role of public administrations in managing informal urbanization (such as promoting participative project during elaboration of MPs) should not be the same as in controlling illegal urbanization. Consequently, the developed contexts need greater anticipation and leadership from the governments, thus reaching a controlled post-legalization scenario.

## 5. Conclusions

The management of the illegal settlement in the SSB is currently at a legal crossroad. The residential occupation of this space has suffered a constant and growing evolution. The response of regional and local governments has evolved, but it still responds to old parameters such as political strategy and pressure from property owners' lobbies.

The regional press shows that the work of technicians and public officials (police, prosecutors, and judges) has been critical in triggering a proposal of solution. Without complaints and sentencing, i.e., without public action in the area of urban discipline, there would currently be no proposal for legalization.

However, this proposal is not enough, and the response of the municipal government still follows an action–reaction policy, as also shown by the press. Regional and local governments have to anticipate and innovate. Their old-fashioned procedures create stagnation without spatial order in both LDL and NDL.

Exploring the third way [45] in the Mediterranean context must ensure the leadership of the governments in order to maintain the criterion of public interest over private. For instance, that means environmental legislation must prevail over urban planning and not only be applied as a punishment. Thus, municipal governments must take the lead and ensure implementation of legalization measures in which the owners assume more responsibility. These requirements do not have to consist of traditional measures such as promoting same urban standards to all urban owners (illegal and legal ones) but rather making each owner responsible for the environmental footprint they generate and for reducing it. Becoming legal must mean becoming sustainable. Illegal settlements in Mediterranean

rural areas must consider their own natural resources and promote their sustainability. This was a common and sustainable practice used in previous dispersed agricultural dwellings in Spain.

In a region with nearly 40,000 dwellings subjected to the same situation [66], the acceptance of this new paradigm is key to placing property rights below the principle of sustainability and ecological transition. Solving the case of this study area under these parameters can be an example and can be used to promote better management practices for this type of settlement in similar contexts.

**Funding:** The research was funded by Academic Insertion Program of Pontifical Catholic University of Chile.

**Conflicts of Interest:** The author declares no conflict of interest. The funders had no role in the design of the study; in the collection, analyses, or interpretation of data; in the writing of the manuscript; or in the decision to publish the results.

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
