# Peer review of "Evolution and Management of Illegal Settlements in Mid-Sized Towns. The Case of Sierra de Santa Bárbara (Plasencia, Spain)"

_sustainability, doi:10.3390/su12083438_

Round 1

Reviewer 1 Report

The paper investigates informal settlements what makes it an interesting and original approach however the aim of the paper is not clearly specified. The reader finds the analyze of the changes in the built-up areas of the undevelopable land together with the description of the legalization attempts. Some more background of the town and its development should be given for the reader who might not be familiar to the city history and present growth. The reasons for the illegality are not clearly marked. Who and why continuous to construct building in the restricted area, what are the benefits and threats, what sort of building are being built, to what extent the buildings are associated to agriculture, how many houses serve as second houses and how many are permanently inhabited, what is the landownership structure of the land, to what extent the housing is related to suburbanization, urban sprawl etc. Those questions should be answered in the paper. The author writes of the necessity of the infrastructure development when the process of the legalization proceeds but on the other hand suggests elimination of the unproductive gardens (private?)  it is not clear how can that goals be gained.     

The abbreviations used and repeated (MP, CNIG, WMS, ESR,RUCI, LDL, SSB, UDL make the paper difficult to read

Figure 1 does not give any relevant information ; the connection to the climate and soils problems if needed should be extended

What does the formal and informal cities means ? – line 74

Lines 112-114 I have chosen should be rather changed to  have been chosen

Plasencia with the population over 30 thousand should be rater included in a medium sized group of towns

Figure 4 Housing dating is not readable

Figure 5 Difficult to read, urban planning classification gives 3 types of land, does that mean that there is no more unbuilt undevelopable land? What is than the reason to call it that?    

Author Response

First of all, I would like to thank the first evaluator for his work and comments. Without a doubt, they have made a great contribution to my work. I have considered most of your suggestions and answered all of them. From my point of view, I have improved the paper (I have added and modified text where it was required). The figures have also been modified and there is a new figure 1.

Next, I explain my work:

⃝The paper investigates informal settlements what makes it an interesting and original approach however the aim of the paper is not clearly specified.

The last part of the introductory section has been modified. The aim of the work is explicitly included. In addition, the wording is simplified.

“The aim of this work is to demonstrate that the strategy of ignoring the problem (first way in urban policies) causes its increase, just as the second way is a specific solution in space and time, activated only in extreme cases for the competent government (conflict social, legal problems or media attention). Therefore, this work evidence the first way is an erroneous urban policy and the second way should only be a good option if it represents a path to the third way in the future (learning and planning improvement).”

The reader finds the analyze of the changes in the built-up areas of the undevelopable land together with the description of the legalization attempts. Some more background of the town and its development should be given for the reader who might not be familiar to the city history and present growth.

A description of historical evolution could blur the purpose of the text. However, its recommendation has been met by incorporating a very brief historical review. Information is included through a new reference and it is compared with Cáceres (capital city of the province):

“Within its closest geographical area, Plasencia has historically occupied an important role. Before the provincial division of Spain in 1833, its demographic differences with Cáceres (current capital city) were minor. In 1797 (new reference), Plasencia had 4,852 inhabitants and Cáceres had 6,860 inhabitants. Since then, the concentration of administrative services in the capital city has prejudice Plasencia, although it has maintained its influence throughout the north of the province.”

Also, a broader geographic framework is provided throughout a new paragraph and new Figure 1:

“Furthermore, the growth of its territorial relevance has been limited by the influence of other provincial capitals (such as Salamanca) or large cities in other provinces (Talavera de la Reina, province of Toledo), as well as by the proximity of the national border between Spain and Portugal.”

Residential developments in the study area (SSB) are included in the paper, from its origin to the present moment.

The reasons for the illegality are not clearly marked. Who and why continuous to construct building in the restricted area, what are the benefits and threats, what sort of building are being built, to what extent the buildings are associated to agriculture, how many houses serve as second houses and how many are permanently inhabited, what is the landownership structure of the land, to what extent the housing is related to suburbanization, urban sprawl etc. Those questions should be answered in the paper.

It is impossible to get to some of the requested data. This urbanization is illegal and clandestine. For this reason, there are no official data. In previous studies I have tried to approach homeowners to get information, but in a strict sense they are people who do not comply with the law and do not want to give personal data. This is a limitation for this type of study.

1---Who and why continuous to construct building in the restricted area:

Now, they are also in the middle of a judicial process with the threat of demolition of their homes. Under these circumstances, they do not provide information about their condition such as first or second residences (no official data on this) or other matters.

2---what are the benefits and threats:

New text has been added:

About the benefits:

“The attractiveness of the study area lies mainly in the possibility of acquiring or building a dwelling within a natural environment but close to an urban center. In addition, they are more spacious and comfortable housing typologies at a lower cost, because the homeowners save the costs associated with legal housing (taxes and legal payments to construction professionals).”

About the threats:

“However, this authorization is difficult because building dwellings within natural areas involves threats. The houses do not have all the services and infrastructures of urban land (garbage collection service, web connection, urban equipment, etc.) and therefore, they are potential sources and victims of socio-natural risks and various types of pollution.”

3---what sort of building are being built:

A precision has been made:

“The average floor plant size of dwellings is 149.45 m2 (69.68% exceeds 90 m2 and 24.86% is above 180 m2), with a very high prevalence, but not specified in this work, of single-storey houses (upper two-storey houses do not exist).”

4---what extent the buildings are associated to agriculture:

Strictly speaking, there are no houses associated with agriculture in the study area. Sometimes small orchards appear next to houses, but without demonstrable continuous activity.

Therefore, agricultural practices are limited to leisure issues. Also, steep slopes and small size of the plots make impossible upper dimension of agricultural activities.

There are two possible references to this in the text (Conclusion section). Both have been specified.

First, I was referring to the old houses scattered in rural areas. I have specified that it is a general question and not focused on the study area.

Secondly, I was referring to the prevalence of agriculture in rural areas. This traditional use can be basic in the economy as it is productive, while residential leisure use must be secondary.

5---how many houses serve as second houses and how many are permanently inhabited:

There is no way to obtain this information, either through official channels (these data do not exist) or by asking the owners directly (they do not accept such questions).

Due to personal knowledge of the area, I believe that there is a predominance of second homes that is decreasing. There is a transformation from second residences to main residences. However, this is only a belief and is not objective data. For this reason, I do not include it in the work.

6---what is the landownership structure of the land:

There is new information about this:

“The land ownership structure is highly fragmented, so the average area of land properties is 1.33 hectares (it does not reach the 1.5 hectares required by law to build a home in NDL). In fact, under these conditions are 85.56% of the plots built in the study area. In fact, judicial boundaries of the plots and those observed in the orthophoto do not coincide, which means that the largest plots have been illegally divided.”

7---what extent the housing is related to suburbanization, urban sprawl:

From my point of view, those concepts require the subjective response of homeowners. Suburbanization is not only a physical and objective concept (increase and dispersion of the urbanized area), but it is related to people's motivation.

Some possible motivations are included in the text, but it is not possible through them to quantify what exact percentage belongs to each urban phenomenon.

The peculiarity of illegal urbanization makes it difficult to achieve such responses. The methodology also does not contemplate the individual part of the owners. That is not the objective of the work. Anyway, previous experiences indicate that illegal owners do not want to answer about their preferences.

The author writes of the necessity of the infrastructure development when the process of the legalization proceeds but on the other hand suggests elimination of the unproductive gardens (private?)  it is not clear how can that goals be gained.     

Within the urban legalization processes in Spain it is common to replace the land property transfer by a monetary exchange (from private to public). This results in the persistence of the damage caused. If the exchange of land ownership (a mechanism used in urban land) is maintained as something common in NDL, the public administration can promote real changes in the physical environment.

Some changes have been made in the text to include this idea. It is a mechanism used in urban land that can serve to achieve the proposed goals:

“Becoming legal must mean becoming sustainable. For instance, this involves preventing the conversion of land property transfers into money transfers.”

“To achieve the latter, it is more important to build permeable pavements, or a reorganization of public and private properties based in land transfer mechanism (this makes possible the transformation of unproductive irrigated areas, such as private gardens, into areas of public infrastructures with sustainability standards).”

The abbreviations used and repeated (MP, CNIG, WMS, ESR,RUCI, LDL, SSB, UDL make the paper difficult to read

The use of MP has been reduced (from 23 to 17). Redundancies have been removed.

Have been totally supressed the use of CNIG, WMS and ESR.

RUCI only appears 6 times in the text. All of them essential.

The use of LDL has been reduced (from 11 to 8).

The use of SSB has been drastically reduced from 41 appearances to 17. Other words have been used (such as “this place”, “study area”, “geographical location”, “area”) or sentences have been rewritten.

The term UDL has been changed to NDL (non-developable land). Its use has been minimized.

Figure 1 does not give any relevant information ; the connection to the climate and soils problems if needed should be extended

Former Figure 1 has been removed to improve the geographical background. New Figure 1 is now located at epigraph 3.1.

I think that the connection between climate and soil is not a central theme of the study. The study focuses on illegal residential development and its management, so environmental issues could be important but not central to this work.

What does the formal and informal cities means ? – line 74

There is a translation mistake. I meant urbanizations not cities. For this reason, the sentence has been rewritten. The new phrase changes the meaning to express what it really meant.

“Dealing with formal urbanizations in legal terms and dealing (or not dealing) with informal and illegal urbanizations evince parallel management (that is, two ways of doing that never meet and cross) [43], but the legalization processes make them clash.”

Lines 112-114 I have chosen should be rather changed to  have been chosen

Done.

Plasencia with the population over 30 thousand should be rater included in a medium sized group of towns

The suggestion has been accepted. This can be verified in the text (first paragraph of section 3.1.), and in the title of the paper.

“The principal urban center of the latter (360-430 m) is a mid-sized town (30,913 inhabitants) located in the central - west side of the Iberian Peninsula.”

“Evolution and management of illegal settlements in mid-sized towns. The case of Sierra de Santa Bárbara (Plasencia, Spain)”

Figure 4 Housing dating is not readable

Figure 4 has been improved.

The font size is larger in both the chart and the legend.

In addition, other improvements have been made:

  • New placement for the labels.
  • New symbology (colours and symbols combined) for better differentiation.
  • Background colour in the layer (LDL) has been changed, for better contrast.
  • Background colour has been added to the chart.

Figure 5 Difficult to read, urban planning classification gives 3 types of land, does that mean that there is no more unbuilt undevelopable land? What is than the reason to call it that?   

This is possibly due to a translation mistake. The correct expression is "non-developable Land" (NDL). This expression has been changed throughout the text and all the figures, not only in #5.

The titles in the legend have been modified.

Spanish legislation establishes 3 basic land classes (there are many subclasses). Therefore, the word "Simplification" is included in the legend of the new figure 5.

It cannot be legally built in NDL, with exceptions (explained in the text).

There are other improvements on the scale, such as adopting the points as decimal separators (previously they were commas). Also, the point size is larger.

Reviewer 2 Report

This paper is potentially a significant contribution to the differentiated spatial dynamics of informality in the global north and the socio-political factors that underpin this process. However, the paper at its present state requires significant improvements, especially in its analytical rigour, the quality of writing and implications for theory and practice. The following comments are therefore made to improve the current state of the paper.

Major comments

The title of the paper is vague and lacks clarity to capture its content.

The abstract needs to be revised. How does the author condition the statement ‘thanks to historical orthophotos series’ in terms of scientific writing?

Introduction

Although the line of argument in the introduction is quite clear, a theoretical or literature review on the subject would fine tune the author(s) reasoning and framework for the analysis. The introduction brings together themes of sustainability, politics and public administration. It is not very clear the author’s line of argument or direction.

There are emergent questions here: how does this form of so-called informal second home housing or illegal construction practices (in the Mediterranean region) position within the broader literature on informality in the global North. The author may consider the works of:

  1. Chiodelli (2019) The Dark Side of Urban Informality in the Global North: Housing Illegality and Organized Crime in Northern Italy

https://onlinelibrary.wiley.com/doi/full/10.1111/1468-2427.12745

  1. Curci (2012): ‘The informal component of mediterranean littoralization: unlawful ricreational homes by the sea at the turn of the third millennium

https://www.politesi.polimi.it/handle/10589/68281

Results 

Since the author appears to utilize media accounts as a source of data to develop his argument on the contestations of illegal housing construction in the study area, the work of Richard Grant could provide a useful reference to develop for the framing the actions and interactions between the actors and stakeholders.

Grant, R (2006) Out of place? Global citizens in local spaces. A Study of the Informal Settlements in the Korle Lagoon Environs in Accra, Ghana. Urban Forum, 17(1).

The author also seems to imply that socio-political factors play a role in the forms of illegal constructions in the results. However, there is no engagement of this at literature or theoretical level.

The results of the study are not properly presented due to the style of writing and lack of an analytical framework to support its discussion. This feeds into the next point.

Conclusions

Even though the author provides conclusions, a discussion of the results would properly underline this paper's contribution to the existing literature in the field (how do the results in this work correspond or contradict existing studies) and eventually planning and policy implications.

Line 366-376. How are these suggestions supported by the results and discussion of findings? I do not find the specific results that provide the basis for this suggestion.

Minor comments

Line 44 ‘including Spain and Plasencia’. Isn’t Plasencia part of Spain?

Line 103 what is ‘legal existing conditions’. Do you mean existing legal conditions?

Line 111 Consider changing section title

Line 177-178 considered an important urban centre at the regional level (4°) and at the provincial level (2°).  What do the numbers mean?

Line 199 ‘secondary ones are dirt’. What does this mean here?

Line 202 soil classes. I am not sure if this is the right terminology to use.

Line 278-282  If this is a recommendation or suggestion, it should be placed in the appropriate section of the paper.

Line 312- 313 although for different reasons though. This line is a bit confusing

There are several sentences that require revising and reframing. I have lost count of them. The author should read the paper thoroughly to identify address such minor errors.

Author Response

First of all, I would like to thank the second evaluator for his work and comments. Clearly the observations have helped me to improve the presentation of results and launch a discussion in a broader framework. I appreciate the clarity when making recommendations and references to specific texts and their identifiers.

This paper is potentially a significant contribution to the differentiated spatial dynamics of informality in the global north and the socio-political factors that underpin this process. However, the paper at its present state requires significant improvements, especially in its analytical rigour, the quality of writing and implications for theory and practice. The following comments are therefore made to improve the current state of the paper.

Next, I explain the changes one by one. I recommend checking the changes through Microsoft Word Change Control Tool.

MAJOR COMMENTS

The title of the paper is vague and lacks clarity to capture its content.

Title have been changed. The new title is more precise about the content:

“Evolution and management of illegal settlements in mid-sized towns. The case of Sierra de Santa Bárbara (Plasencia, Spain)”

The abstract needs to be revised. How does the author condition the statement ‘thanks to historical orthophotos series’ in terms of scientific writing?

 That expression is a result of mistranslation.

The abstract has been revised:

“The illegal urbanization of rural areas near cities has unveiled failures in urban management. In many cases, urban policies have ignored this fact until the spaces have consolidated. This is the example of the Sierra de Santa Bárbara (Plasencia, Spain), where legalization becomes one of the most feasible solutions. The present work analyses its residential evolution during the last four decades through historical orthophotos review. Along with this, it evaluates public-private conflicts (homeowners vs municipal government) using regional newspaper archives. The results indicate that the strategy of ignoring illegal development increases these problems, leading to legalization as the only possible urban policy. In conclusion, the administration's response is delayed and forced by critical consequences, which prevents learning in urban policies and new solutions that join legality and sustainability.”

Introduction

Although the line of argument in the introduction is quite clear, a theoretical or literature review on the subject would fine tune the author(s) reasoning and framework for the analysis. The introduction brings together themes of sustainability, politics and public administration. It is not very clear the author’s line of argument or direction.

There are several changes in the text (21 new references included in the paper). The main line of the text is intended to be public administration and urban policies. Despite this, it is important to introduce that there is a legal justification for the actions based on sustainability terms. At least, that should be a common goal for Administrations.

Anyway, there are still difficulties in finding bibliography related to this illegal urbanization (as Chiodelli -2019- said), since most of the bibliography on informal settlements deals with studying degraded urban areas and / or in developing or, even, developed countries.

I have enriched this section with ideas from 9 other works (apart from the recommended ones below). Please review the text with the Microsoft Word Change Control Tool.

There are emergent questions here: how does this form of so-called informal second home housing or illegal construction practices (in the Mediterranean region) position within the broader literature on informality in the global North. The author may consider the works of: 

  1. Chiodelli (2019) The Dark Side of Urban Informality in the Global North: Housing Illegality and Organized Crime in Northern Italy

https://onlinelibrary.wiley.com/doi/full/10.1111/1468-2427.12745 

  1. Curci (2012): ‘The informal component of mediterranean littoralization: unlawful ricreational homes by the sea at the turn of the third millennium

https://www.politesi.polimi.it/handle/10589/68281

I have followed the reviewer's recommendations.  I have also added 4 new texts to make a conceptual clarification (Alfaro et al., 2018; Zeković, Petovar and Saman, 2020; Kacerauskas, 2018; De Biase and Losco, 2017). The difference between informal settlements and illegal settlements is explained. At least, its use in this work. Through this, the peculiarity of the Mediterranean basin is shown.

It has been impossible to access Curci's text (only available upon request from users of the institution). However, I have followed the author and I have found an interesting text by Curci about the role of second homes. It is a new reference.

Results 

Since the author appears to utilize media accounts as a source of data to develop his argument on the contestations of illegal housing construction in the study area, the work of Richard Grant could provide a useful reference to develop for the framing the actions and interactions between the actors and stakeholders.

Grant, R (2006) Out of place? Global citizens in local spaces. A Study of the Informal Settlements in the Korle Lagoon Environs in Accra, Ghana. Urban Forum, 17(1).

I understand the reviewer's concern about this type of methodology and the role of the press and its potential subjectivity. It is not mentioned in the text but only a small selection of search results is displayed (almost 3 dozen out of a total of 2 hundred news - published between the 90s and today-). To escape from potential subjectivity, two newspapers are used (each one belongs to a different editorial group, one being more conservative and the other more progressive).

In any case, I believe that the press does not have a subjective role or qualify the situation as in Grant's text (Grant, 2006, p. 2): “The media describe the settlement as "out of place," "a no-man's land" as well as "a hideout for armed robbers, prostitutes, drug pushers and all kinds of squatters"”.

The literal statements are always shown (in quotation marks) and when a description of the positions is made, it is supported by different news. In this case, the press does not take sides.

This methodology is used due to the impossibility of accessing the declarations of the owners and administration through interviews. In addition, it provides a historical vision of the process.

Another point that limits this methodological process is that this case is not a “ended case” like that Accra´s case. So, I could not say something like this (Grant, 2006, p. 15): “The court's final decision was interpreted as a legal victory for government but a political victory for the community. International pressure was then placed on the government not to evict the settlers without considering alternatives or without consulting the community”.

The author also seems to imply that socio-political factors play a role in the forms of illegal constructions in the results. However, there is no engagement of this at literature or theoretical level.

There is a new treatment of this topic in the introduction (conceptual differentiation between informal settlements and illegal settlements) that is developed later in the new section (discussion). In it I have used examples from the literature that you recommended, among others.

The results of the study are not properly presented due to the style of writing and lack of an analytical framework to support its discussion. This feeds into the next point.

I have modified the results section. I have added more precise comments. A new discussion section is included.

Conclusions

Even though the author provides conclusions, a discussion of the results would properly underline this paper's contribution to the existing literature in the field (how do the results in this work correspond or contradict existing studies) and eventually planning and policy implications.

A new discussion section is included. I wanted to highlight the difference between managing illegal urbanization and informal urbanization. I think that literature provides general solutions without considering that there are other cases with particular characteristics (“outlaw settlements” with middle class homeowners). This is the main criticism.

Line 366-376. How are these suggestions supported by the results and discussion of findings? I do not find the specific results that provide the basis for this suggestion.

The entire paragraph has been deleted. It provided very specific solutions not supported by the results obtained. They were recommendations based on previous experiences and therefore do not make sense in this study.

MINOR COMMENTS

Line 44 ‘including Spain and Plasencia’. Isn’t Plasencia part of Spain?

Yes, you are right. It was redundant. It was used to emphasize that Plasencia was within that context. It is corrected (Plasencia was removed from the sentence).

Line 103 what is ‘legal existing conditions’. Do you mean existing legal conditions?

Yes, you are right. There is a translation mistake. It is corrected.

Line 111 Consider changing section title

The section title has been changed:

“2.1. Evolution and characterization of the study area”

Line 177-178 considered an important urban centre at the regional level (4°) and at the provincial level (2°).  What do the numbers mean?

This has been clarified. We have redrafted the phrase:

“Despite its size, Plasencia is the fourth most populous city at the regional level and the second one at the provincial level.”

Line 199 ‘secondary ones are dirt’. What does this mean here?

It is a translation mistake. I wanted to express that the main roads are paved but the secondary ones are not (they are dirt or sand tracks). It is corrected:

“Land plots are irregular and atomized, delimited by metal fences and roads (main ones are paved, but secondary ones are not).

Line 202 soil classes. I am not sure if this is the right terminology to use.

You are right. It is a translation error. The correct expression is “land”. It has been changed throughout the text.

Line 278-282  If this is a recommendation or suggestion, it should be placed in the appropriate section of the paper.

It is not a recommendation or suggestion. This is a legal fact. In any case, the phrase changes to improve understanding:

“Without regional approval, action is not viable. According to law, an environmental report must be approved by regional government in order to modify the existing MP.”

Line 312- 313 although for different reasons though. This line is a bit confusing

I wanted to express that there are projects that turn enemies into allies, such as the implantation of a wind farm. Everyone rejects it, from the municipal government to illegal homeowners and environmental associations. This is paradoxical because homes are already generating the same impact. The sentence has been rewritten:

“Nevertheless, there are projects such as the implementation of a wind farm that produces a common negative answer by all parts for its visual, noise and environmental impact in the area”.

There are several sentences that require revising and reframing. I have lost count of them. The author should read the paper thoroughly to identify address such minor errors.

An in-depth review of these issues has been conducted. In any case, if this continues to be a problem, I declare my decision to request the editorial assistance offered by the Sustainability Journal.

Round 2

Reviewer 2 Report

The author(s) has, to a large extent, addressed the major issues in the original manuscript. This revised version is, therefore, a major improvement on the earlier version. I have a small concern with the style of writing but it does not necessarily weaken the arguments, reasoning and conclusions. Going forward, the author(s) should pay attention to these future outputs.